# Are “Free From” Foods Risk-Free? Lactose-Free Milk Fermentation Modulates Normal Colon in a Gut Microbiota in Vitro Model

**DOI:** 10.3390/microorganisms13092021

**Published:** 2025-08-29

**Authors:** Flavia Casciano, Lorenzo Nissen, Alessandra Bordoni, Andrea Gianotti

**Affiliations:** 1DiSTAL—Department of Agricultural and Food Sciences, *Alma Mater Studiorum*—University of Bologna, P.za G. Goidanich, 60, 47521 Cesena, Italy; cascianoflavia@yahoo.it (F.C.); alessandra.bordoni@unibo.it (A.B.); 2CIRI—Interdepartmental Centre of Agri-Food Industrial Research, *Alma Mater Studiorum*—University of Bologna, P.za G. Goidanich, 60, 47521 Cesena, Italy

**Keywords:** dairy, gut microbiota, health halo, *Bifidobacterium* spp., *Enterobacteriaceae*

## Abstract

Nowadays, the consumption of “free from” foods by non-specific consumers is increasing, partly due to a misperception of labels that make them seem healthier. These foods are formulated for consumers with allergies or diseases that limit their diet, and it is not known if there are more benefits than risks for healthy consumers. For example, there is no work investigating the interaction between lactose-free milk and the colonic microbiome of healthy individuals. To focus on the potential modulation of gut microbiota of healthy subjects by lactose-free milk, we performed an in vitro simulation of digestion and fermentation, integrating microbiomics and metabolomics approaches to study changes in gut microbiota populations and metabolite production. Results indicated that lactose-free and lactose-containing milk differently modulated colonic microbiota based on several microbiological indicators, including the reduction in *Bifidobacteriaceae* (approximately more than two times) and *Lactobacillales* and the reduction in the beneficial production of microbial compounds (approximately six times less acetic acid and two times less butanoic acid). Such features suggest that lactose-free milk increases the risk of dysbiosis in healthy subjects. Our work identifies the drivers of this dysbiosis among hundreds of molecules and microbes of the gut microbiota, assigning specific names and ecological niches for the first time. It employs an in vitro model, which represents a new standard for sustainable research and improves translatability. Our findings support the European Society for Clinical Nutrition and Metabolism (ESPEN) guidelines, which do not recommend the routine consumption of lactose-free diets in the absence of diagnosed intolerance.

## 1. Introduction

The disaccharide lactose, the dominant carbohydrate in dairy products, is hydrolyzed by the lactase enzyme, which is abundant in the proximal jejunum and decreases progressively towards the ileum [1]. The resulting galactose and glucose are then actively transferred into the bloodstream. Lactose that is not digested arrives in the colon where it is broken down into monosaccharides by gut microbiota and fermented to produce gases and short-chain fatty acids. The absence or deficiency of lactase, commonly related to aging, is the main cause of lactose intolerance (LI) [2,3,4]. Excess undigested lactose draws water from the blood vessels into the gut lumen, causing loose stools or watery diarrhea [5]. The gases produced during bacterial fermentation increase pressure in the large intestine, leading to gut symptoms including flatulence, bloating, and various types of abdominal pain [6].

The prevention of gastrointestinal symptoms of LI is based on a reduction or elimination of lactose from the diet. To enable LI people to consume milk and dairy products, which provide essential macro- and micronutrients, “low lactose” or “lactose-free” (LF) products are industrially produced [7].

Consumption of LF products by people who are not intolerant is increasing for several reasons: (i) families switch completely to LF foods although only one member is intolerant, (ii) LI is often self-diagnosed [8], and (iii) there is an increasing negative attitude towards milk-derived foods due to a perceived risk associated with lactose intake [9]. “Free-from” diets are a new trend. In supermarkets, a wide range of products labeled “lactose-free” is now available. The global LF dairy market size was valued at USD 11.45 billion in 2021 and is projected to reach USD 24.36 billion by 2031, growing at a compound annual growth rate (CAGR) of 8% from 2022 to 2031 [10]. Such value is not only due to the increasing number of diagnoses of lactose intolerance but also due to the rising consumption by non-specific consumers.

Additional evidence shows that for the development of intestinal eubiosis, a lactose-free diet is detrimental in infants [11]. Currently, under normal conditions, the benefits to the gut microbiota of consuming lactose from dairy products are well established [12,13,14]; otherwise, the impact of consuming LF products remains unclear. From this perspective, our study aims to highlight possible variations in the colon microbiota after the ingestion of lactose-free milk, identifying the possible culprits by focusing on microbial species and microbial metabolites. We hypothesize that the removal of lactose from dairy products may alter gut microbiota composition in healthy individuals by reducing the availability of lactose-utilizing beneficial bacteria (e.g., *Bifidobacterium*, *Lactobacillus*) while potentially favoring the expansion of opportunistic taxa such as *Proteobacteria*, which can exploit alternative nutrient niches.

To study the impact of LF milk on the colon microbiota of lactose-tolerant donors, we sequentially applied the protocols of INFOGEST in vitro digestion [15] and MICODE (Multi-Unit In Vitro Colon Model) [16] to colonic fermentation charged with human colon microbiota (HCM) from lactose-tolerant adult donors in this work. Modulations of microbiota populations and the production of metabolites were assessed by means of omics and multivariate statistics. Additionally, results were compared to a previous work conducted on HCM of LI subjects [17].

## 2. Materials and Methods

### 2.1. Milk Samples

UHT (Ultra-High-Temperature) semi-skimmed milk (L) and UHT semi-skimmed lactose-free milk (LF) (Granarolo S.p.A., Bologna, Italy) were purchased at a local market. The lactose concentration in LF was < 0.1 g/L, as declared by the supplier.

### 2.2. Experimental Workflow

By processing milk samples through gastro-duodenal digestion using the INFOGEST protocol [15] and then transferring the digestates to the MICODE in vitro colon model with human colon microbiota (HCM) [16], we simulate human proximal colonic fermentation. This approach allowed us to observe the shifts in the colon microbiota and its metabolites during fermentation, for a comprehensive way to understand how milk components are digested and fermented in the human gut.

### 2.3. Human Colon Microbiota

Human colon microbiota (HCM) were obtained from the stools of two healthy donors (one male and one female) aged between 30 and 45 y, respectively. The methods used for the selection of donors using inclusion criteria and the protocols for stool collection have been previously published [17,18,19,20]. Briefly, donors were omnivorous, not smokers, not overweight, and did not consume antibiotics, pre- or probiotics three months before donations. HCM was prepared by mixing 2 g of each stool in 36 mL of pre-reduced phosphate-buffered saline (PBS) [21,22,23]; the mixture was subsequently washed twice with PBS (6 min at 160,000× *g*). Donations were obtained two different times from the same two donors, in order to replicate the experiment.

### 2.4. In Vitro Intestinal Model

The gut model was created by combining the INFOGEST method [15] for oro-gastro-duodenal digestion and MICODE model [16,17,19] for colonic fermentation. Milk samples were in vitro digested in triplicate, and a blank digestion without any food was also performed. At the end of the duodenal phase of vitro digestion, digestates were collected and kept at −80 °C. Triplicates of L or LF digestion were homogeneously combined and then inoculated in MICODE bioreactors, as reported previously [17,21,23]. Twenty-four-hour proximal colonic fermentations were carried out in separate vessels, following published protocols [17,24]. The full procedure has been previously described [19]. Once exact ecological settings were obtained, three different bioreactors were added with 9 mL of HCM suspension and either (i) 1 mL of digested LF; (ii) 1 mL of digested L; or (iii) 1 mL of blank control (BC) of digestion. Sampling was performed at the baseline (BL) and after 16 h (T1) and after 24 h, i.e., at endpoint (EP) of fermentation. The BL (i.e., the adaptation of microbiota to in vitro condition) was obtained at 2.26 ± 0.12 h, defined by the first acidification of the medium read by the integrated software Lucullus 3.1 (1 read/10 s) (Securecell AG, Urdorf, Switzerland) which also keeps a record of all settings during experiments. Fermentations were conducted two times using two different pools of stools from the same donors. Time points were defined as previously described [17,18,19,20,21,23]; for example, a 24 h time point is taken to anticipate the beginning of the stationary phase of growth and avoid microbial growth inhibition by toxic microbial metabolites in a batch-controlled system. At each time point, aseptic sampling of 4 mL from the volume of each vessel was performed and this volume was centrifuged at 16,000× *g* for 7 min to separate pellets from the supernatants. The former were used for microbiomics and the latter for metabolomics. The pellets were washed twice in O_2_-reduced PBS to remove stool debris and were used to extract bacterial DNA. Bacterial DNA and supernatants for metabolite profiling were stocked at −80 °C.

### 2.5. Metabolomics

#### 2.5.1. Volatilome Analysis

The profiles of volatile organic compounds (VOCs) were obtained with an Intuvo Agilent 7890A Gas Chromatograph (Agilent Technologies, Santa Clara, CA, USA) equipped with a Chrompack CP-Wax 52 CB capillary column (50 m length, 0.32 mm ID) (Chrompack, Middelburg, The Netherlands). SPME–GC-MS (solid-phase microextraction–gas chromatography–mass spectrophotometry) analysis and data processing were performed following previously published protocols [17,21]. The identification of VOCs was made according to and following the syntax of the NIST (National Institute of Standards and Technology, Gaithersburg, MD, USA) 11 MS Library.

#### 2.5.2. Quantification of Main Microbial VOCs

The key bacterial metabolites associated with fermentation of foodstuffs were measured at the BL in mg/kg by SPME GC-MS, employing a standard and specific cutoffs (LOQ = 0.03 mg/kg and LOD = 0.01 mg/kg) [17]. The values from T1 and EP time points were assessed with respect to BL values as changes. Data were calculated in this order: (i) normalization of the dataset of each single VOC using the mean centering method, (ii) subtraction of the values of BL dataset to the values of dataset of fermentation time points, (iii) generation of ANOVA models, (iv) each VOC was compared between samples by Tukey’s post hoc analysis, (v) representation by box-plots.

### 2.6. Microbiomics

#### 2.6.1. Metataxonomy

DNA samples were extracted using a Purelink Microbiome DNA Purification Kit (Invitrogen, Thermo Fisher Scientific, Carlsbad, CA, USA). DNA samples of BL and EP were used for metataxonomic analysis by 16S rRNA MiSeq sequencing (Illumina Inc., San Diego, CA, USA). Microbiota diversity was obtained by library building and sequencing of 16S r-RNA gene. Libraries were obtained using a MiSeq (Illumina Inc., USA) with paired-end sequencing and a 300 bp read length [25]. Sequences were examined using QIIME 2.0 [26]. Sequencing was commissioned to IGA Technology Service Srl (Udine, Italy).

#### 2.6.2. Quantitation of Bacterial Groups by qPCR

The shifts in quantity, expressed as Log_2_(F/C) [27,28], were evaluated by qPCR and SYBR Green I chemistry [29,30,31] for the following bacterial taxa: *Eubacteria*, *Firmicutes*, *Bacteroidetes*, *Lactobacillales*, *Bifidobacteriaceae*, and *Enterobacteriaceae* [21].

### 2.7. Data Mining and Statistics

Datasets for metabolomics were processed for normality and homoscedasticity [32] using one-way ANOVA (*p* < 0.05), Principal Component Analysis (PCA), and multivariate ANOVA (MANOVA). Datasets for microbiomics were processed to obtain alpha bio-diversities from BIOME files of MiSeq analyses and beta bio-diversities as PCoA (Principal Coordinate of Analysis) using the EMPeror tool [33] from QIIME 2. The dataset for metataxonomy underwent ANOVA for group comparison (BL/EPs) and significant variables (*p* < 0.05) were picked to calculate the shifts in abundance as Log_2_(F/C) and a post hoc Tukey HSD test (*p* < 0.05) was applied. The Multiple List Comparator tool (https://molbiotools.com, last accessed on 27 June 2025) served to generate pairwise intersection maps and a Venn diagram. Log_2_(F/C) results of species level were visualized with Volcano plots, using VolcaNoseR [34]. The dataset from qPCR values was computed for MANOVA and Tukey’s post hoc test. Shifts in qPCR values are presented as Log_2_(F/C) and prepared with BoxPlotR [35]. Normalization of datasets was performed using the mean centering method. Statistics and graphics were made with Statistica v.8.0 (Tibco, Palo Alto, CA, USA).

## 3. Results

### 3.1. Metabolomics

#### 3.1.1. Volatilome Analysis

From the 18 duplicated profiles of SPME GC-MS, 80 molecules with at least 70% similarity were identified using the NIST 11 MSMS library (NIST, Gaithersburg, MD, USA) (Figure 1a). A PCA of 11 organic acids clearly discriminated samples based on colonic fermentation time and sample type (Figure 1b). Butanoic acid was the main descriptor of gut fermented LF (approximately 35.80% of total production) (Appendix A). At EP, pentanoic acid, propanoic acid, 2-methyl were the main descriptors of L (approx. 52.02% and 65.48% of total production, respectively) (Appendix A). Benzoic acid, methyl ester, and octanoic acid were produced only after L fermentation. The production of alcohols depended on fermentation time, and discriminated BC from two food matrices which, however, were not distinguishable from each other at any time (Figure 1c). The main descriptors of L were 2-Octen-1-ol, (E) and 1-Propanol (62.75% and 48.33% of total production, respectively), while LF was described by 3-Buten-1-ol, 3-methyl-, benzyl alcohol, phenethyl alcohol, and phenol, 4-methyl (66.9% 42.82%, 51.20%, and 33.23% of total production, respectively) (Appendix A). All these molecules, except 1-Propanol and Phenol, 4-methyl, were absent at BL (Appendix A). A PCA of 11 other VOCs discriminated against samples based on fermentation time rather than matrix (Figure 1d). L was described by dimethyl trisulfide (67.69%), while LF was described by indole (approximately 38.46%) (Appendix A), which was present at physiological concentration (12.29%) at BL and increased throughout fermentation (33.18% and 54.53% at T1 and EP, respectively) (Appendix A).

#### 3.1.2. Short-Chain Fatty Acids

Due to reported positive health effects [36,37,38,39], short-chain fatty acids (SCFA) were quantified at BL, T1, and EP (Appendix A). Normalized values showed that acetic acid was significantly produced by L fermentation (*p* < 0.05) (Figure 2a), shifting the SCFAs ratio to 64:20:16 (acetic/propanoic/butanoic acids), which is close to the optimum ratio 60:20:20, considered an indicator of microbiota eubiosis [40]. In contrast, LF caused an imbalance in SCFA production, leading to an approximately 31:23:47 SCFA ratio.

#### 3.1.3. Indoles and Phenols

As key detrimental VOCs, we selected indole, 1H-indole, and 3-methyl (skatole), the main dead-end products of intestinal bacteria [41], and phenol, 4-methyl (p-cresol), which can activate DNA methylation and modify the cell cycle by reducing colonocyte proliferation [42]. The three detrimental VOCS were quantified at BL, T1, and EP (Appendix A). Normalized values evidenced that fermentation of LF increased indole production, which was reduced after L fermentation (LF vs. L = *p* < 0.05) (Figure 2b).

### 3.2. Microbiomics

#### 3.2.1. Ecological Biodiversity of Colonic Fermentations

Microbiota diversity indices were affected by both L and LF milk, which perturbated colonic microbial population in terms of stability during fermentation and richness of microbiota composition (Figure 3). Alpha diversity indices included richness, measured using the Chao1 index; entropy, measured using the Shannon index; and abundancy assessed by Observed OTUs. Beta Diversity was instead measured using Bray–Curtis Principal Coordinates Analysis (PCoA). Regardless of the type of milk, richness (Figure 3a) and abundancy (Figure 3b) were significantly reduced after fermentation. Entropy was reduced as well, with a significant difference between milk samples (*p* < 0.05) (Figure 3c). Reductions in these indicators are typical in similar experiments when a fermentative substrate has no prebiotic values [21]. Regarding beta diversity, Bray–Curtis PCoA (Figure 3d) indicated a clear time-dependent modulation effect. Furthermore, after colonic fermentations of different samples, microbiota was lodged in three different spatial areas of the graphic, demonstrating that modulation was sample-dependent.

#### 3.2.2. Metataxonomy of Colonic Fermentations

Three different datasets representing taxa abundance at phylum, family, and species levels were prepared. Complete R models of ANOVA of phylum, family, and species levels for MiSeq analysis are reported in the Supplemental Materials. From the larger datasets, OTUs biologically involved in digestion of lactose and dairy products were selected as variables to focus the discussion. In particular, 9 variables were selected for the phylum level (Figure 4a, Appendix A), 21 for family level (Figure 4b, Appendix A), and 25 for species level (Figure 4c, Appendix A). To identify shared taxa between the beginning and the end of fermentations, species-level data were also analyzed using cut-off variables in a Venn diagram (Figure 4c, Appendix A) and pairwise intersections map (Figure 4d, Appendix A). To obtain significances in terms of −Log_10_(*p*) for Log_2_ Fold Changes (Log_2_(F/C)) at family and species levels for Volcano plots (Figure 4f,g and Appendix A), *p* values from ANOVA models were used. At the phylum level (Figure 4a and Appendix A), L and LF similarly reduced the abundances of *Firmicutes* and *Bacteroidetes*. Nevertheless, LF fermentation promoted a higher abundance of *Proteobacteria*, a group that includes Gram-negative pathogens, than L (Appendix A). At the family level, LF fermentation reduced the abundance of some commensal taxa, namely fibrolytic *Bacteroidaceae* and butyrate-producing *Ruminococcaceae* (Figure 4b,f and Appendix A). Both L and LF fermentations mildly modulated commensal *Enterobacteriaceaceae*. L and LF fermentations had opposite effects on *Veillonellaceae* (Figure 4f), which were fostered by LF and reduced by fermentation (Appendix A).

LF and L colon fermentations did not significantly modulate lactic acid bacteria (*Enterococcaceae*, *Lactobacillaceae*, *Streptococcaceae*) specialized for the fermentation of dairy sugars. Notably, bifidogenic activity, measured as the *Bifidobacteriaceae*-to-*Enterobacteriaceae* ratio, followed this trend: BL (2.42) > L (0.24) > LF (0.08) > BC (0.04) (Appendix A). At the end of fermentation, L shared most of taxa found at BL and showed a slightly higher number of exclusive taxa than LF (Figure 4c,d). Among exclusive taxa (Appendix A), L was characterized by two important butyrate producers, *Ruminococcus* and *Faecalibacterium*, whereas LF was characterized by *Bacteroides fragilis.* At the species level (Figure 4e,g, and Appendix A), LF fermentation significantly increased the abundance of *Escherichia* spp. to a greater extent than L fermentation. In terms of beneficial taxa, L fostered *Bifidobacterium bifidum*, probably due to its beta-galactosidase activity.

#### 3.2.3. Enumeration of Selected Bacterial Targets

The shifts observed during fermentation time are reported as Log_2_(F/C) values, where F/C is the ratio of time point/baseline (Figure 5). Bacterial enumeration at the BL, values of single time points, and statistics related to shifts are reported in Appendix A. Both L and LF fermentations decreased the abundance at BL of *Eubacteria* (2.24 × 10^9^ ± 7.00 × 10^7^ cells/mL), *Firmicutes* (2.04 × 10^9^ ± 1.57 × 10^7^ cells/mL), and *Bacteroidetes* (1.47 × 10^8^ ± 1.00 × 10^7^ cells/mL), but with higher significance for LF fermentation (with *p* < 0.05 for *Bacteroidetes*). As for beneficial bacteria, an opposite trend was observed. LF fermentation decreased and L fermentation increased the abundance at BL of *Lactobacillales* (7.86 × 10^4^ ± 4.74 × 10^3^ cells/mL) and *Bifidobacteriaceae* (6.15 × 10^5^ ± 1.64 × 10^4^ cells/mL) (Figure 5). Considering opportunistic taxa, LF fermentation increased the abundance of *Enterobacteriaceae* at BL (2.38 × 10^5^ ± 7.60 × 10^3^ cells/mL) 3.5 times more than L fermentation (*p* < 0.05).

## 4. Discussion

Lactose, a specific component of mammalian milk, is not fully metabolized and absorbed in the jejunum, and a portion of dietary lactose may reach the large intestine [43], where it impacts the composition and metabolome of gut microbiota [1,44]. Previous studies have indicated a positive correlation between lactose consumption and the abundance of *Bifidobacterium* and *Lactobacillus* in adult [45] and infant [46] fecal samples. The exclusion of lactose from the diet of intolerant subjects is mandatory, but its consequences on the gut microbiota of tolerant subjects are often overlooked. To further clarify this aspect, the present in vitro study evaluated the effects of LF milk on the gut microbiota of lactose-tolerant subjects, compared to control lactose-containing milk (L).

Our results demonstrated that LF and L milk differentially modulate the colonic microbiota of lactose-tolerant subjects, with several microbiological indicators suggesting that LF milk increases the risk of dysbiosis in these subjects. Firstly, although colonic fermentation of both milks caused a reduction in alpha biodiversity related to entropy of microbiota, this reduction was greater after LF fermentation. Secondly, LF fermentation decreased acetate and butyrate production, concomitant with a reduction in beneficial *Bifidobacteriaceae* and *Lactobacillales*. Thirdly, LF fermentation was associated with higher levels of indole and overrepresentation of *Escherichia* spp., indicating a potentially harmful scenario for the host. In fact, indole can be toxic to the mucosa and is produced as a tryptophan catabolite by many *Escherichia* species, including pathobionts [47]. Lastly, LF fermentation induced overrepresentation of *Veillonellaceae*, a pro-inflammatory family [48], and the exclusive growth of *Bacteroides fragilis* and *Fusobacterium gonidiaformans*, two potential pathobionts [49,50].

In contrast, L fermentation produced positive effects. It caused a reduction in *Firmicutes* and an increase in *Lactobacillales*, indicating selective effects such as the inhibition of opportunistic populations and promotion of beneficial *Lactobacillales*. L fermentation also resulted in overrepresentation of beneficial *Bifidobacteriaceae* and *Bifidobacterium bifidum*, which is consistent with higher production of health-related SCFAs and medium chain fatty acids (MCFAs), in particular acetic acid and octanoic acid. Similarly, other researchers [36], using a similar in vitro model, demonstrated that *Bifidobacterium spp*. is associated with high levels of acetic acid and 2-Propanoic, methyl acid, whereas *Lactobacillus* and *Enterococcus spp*. are associated with Octanoic acid [51]. These compounds are generally produced during dairy fermentation, particularly from lactose degradation by lactic acid bacteria [51]. They have been linked to important functional properties, benefiting multiple human organs, from the gut to the brain. It is also important to note that overrepresentation of these bacterial groups positively impacts the gut environment and contributes to host health. Our results confirmed that normal microbiota is more prone to dysbiosis when lactose is absent from milk. When the results of the present work are compared to those previously obtained in MICODE with colon microbiota from lactose intolerant subjects [17], some differences are evident (Table 1). LF fermentation by the colonic microbiota of LI subjects showed beneficial effects, including increased positive metabolites, reduction in some detrimental VOCs, and decreased *Peptostreptococcaceae*. In contrast, LF fermentation by the gut microbiota of healthy subjects increased indole production and the abundance of potentially harmful *Peptostreptococcaceae*.

## 5. Conclusions

The consumption of foods with the “free from” attribute is constantly increasing (lactose-free, gluten-free, etc.) although it is not recommended for healthy individuals [52]. These foods are tailored to specific consumers and their formulations and processing differ from normal products. Erroneously, the “free-from” symbols influence consumers’ perceptions of food products and the absence of an ingredient is perceived as a sign of improved healthiness or quality [53]. To date, the consequences of this “health halo effect” are rarely considered or studied, even though no one has explicitly ruled out negative consequences linked to the consumption of these tailored foods by non-specific consumers. 

In this study, a negative modulation of lactose-tolerant gut microbiota through the fermentation of LF milk was reported, suggesting the functional role of the disaccharide in healthy individuals and possible concerns related to its exclusion. Our results do not consider the adaptive mechanisms that might occur during prolonged intake of LF milk in normal subjects. In lactose-intolerant individuals, colonic microbes adapt to the presence of lactose in the colon lumen, sometimes leading to milder and less severe gastrointestinal symptoms. Adaptation could also occur in the opposite situation, and further studies are needed to evaluate this aspect and to validate our findings in vivo. The results obtained with L fermentation may appear exaggerated, considering that the INFOGEST in vitro digestion protocol does not include lactase. Consequently, in our in vitro model, a higher level of lactose reached the colon than would occur in vivo. However, our intent was to focus on the gut microbiota and not on bio-accessibility. From this perspective, the microbiota of normal subjects used in MICODE contained several species that naturally express lactase at levels sufficient to compensate for the absence of lactase in the INFOGEST digestion system. Considering these limitations, the results obtained from the MICODE in vitro model could be valuable for understanding the effects of healthy microbiota interacting with foods tailored for altered microbiota, demonstrating that what is healthy for one is not necessarily healthy for all. This study brings to light the finding that self-made diet restrictions could be harmful in those consumers who do not need them and supports European Society for Clinical Nutrition and Metabolism (ESPEN) guidelines, which do not recommend the routine adherence to lactose-free diets if no intolerance is diagnosed [54].

## Figures and Tables

**Figure 1 microorganisms-13-02021-f001:**
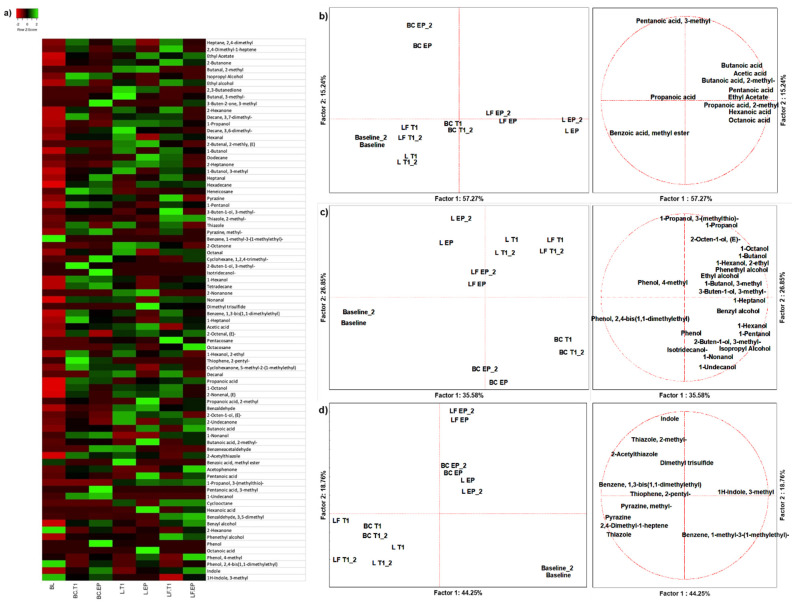
PCA plots of the volatilome sorted by chemical classes. (**a**) Heatmap of complete volatilome; (**b**) acids; (**c**) alcohols; (**d**) other VOCs. (**a**–**c**) Left side diagrams are for PCAs of cases; right side diagrams are for PCAs of variables. L = standard milk; LF = lactose-free milk; BC = blank control; BL = baseline; T1 = 16 h; EP = 24 h.

**Figure 2 microorganisms-13-02021-f002:**
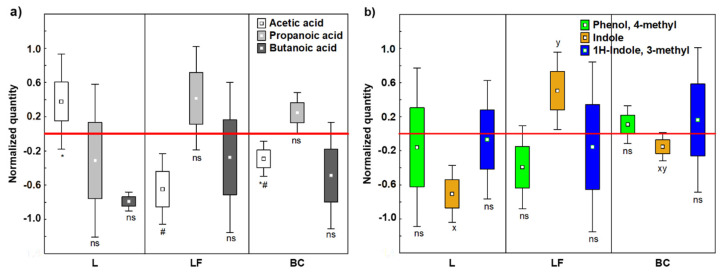
Changes in abundance of (**a**) beneficial microbial VOCs metabolites and (**b**) detrimental microbial VOCs, expressed as normalized scale from relative abundances with respect to baseline of in vitro fermentation (red line). Box plots include all replicas of T1 (16 h) and EP (24 h) values. Marker = mean; box = mean ± standard error; whiskers = mean ± standard deviation. Different letters (x, y) or symbols (#, *) inside the graphs among a single independent variable indicate significant difference according to MANOVA model followed by post hoc Tukey’s HSD test. ns = not significant; L = standard milk; LF = lactose-free milk.

**Figure 3 microorganisms-13-02021-f003:**
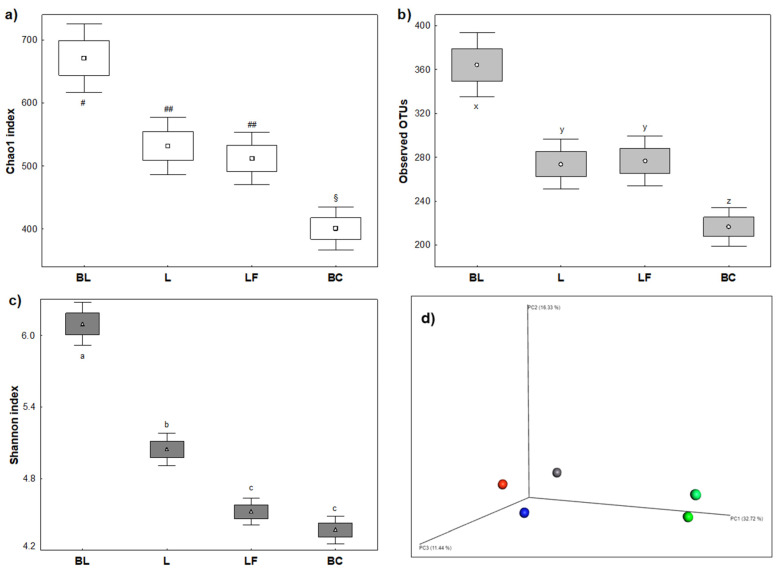
Ecological diversities representing baseline (BL) and end points of colonic fermentation of human colon microbiota. Values were recorded after in vitro digestion and fermentation of different milk samples. (**a**) Chao1 index representing abundance. (**b**) Observed OTUs representing richness. (**c**) Shannon index representing evenness. (**d**) Bray–Curtis PCoA of Beta Diversity representing differences among samples. BL = baseline mean; L = standard milk; LF = lactose-free milk; BC = blank control. Different letters (a, b, c, x, y, z) or symbols (§, #) inside the graphs indicate statistical significance by Tuckey’s post hoc test (*p <* 0.05). Green spheres = BL values; blue sphere = L; red sphere = LF; gray sphere = BC.

**Figure 4 microorganisms-13-02021-f004:**
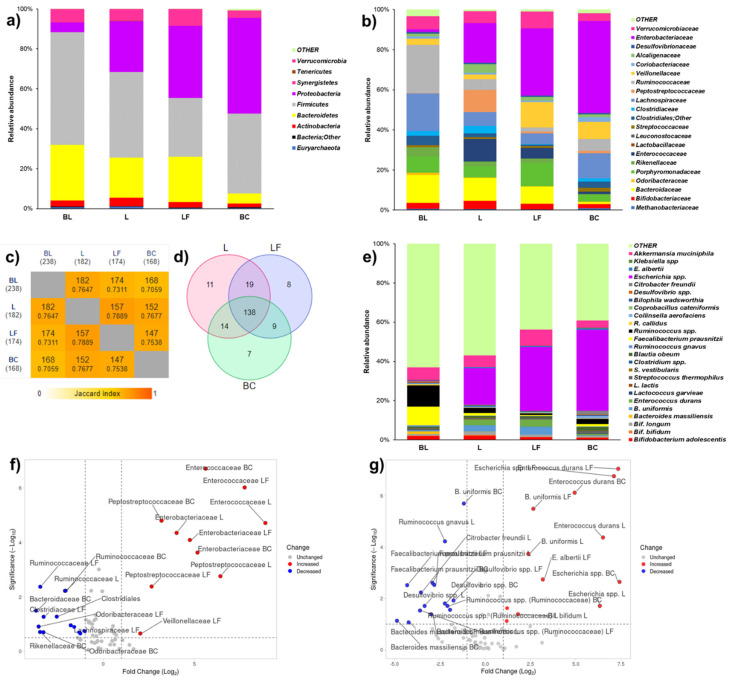
Microbiota 16S-rRNA analyses of baseline (BL) and samples after colonic fermentation: (**a**) metataxonomy and relative abundances at phylum level, (**b**) metataxonomy and relative abundances at family level, (**c**) pairwise intersection map at species level, (**d**) Venn diagram at species level, (**e**) metataxonomy and relative abundances at species level of selected targets, (**f**) volcano plots to indicate changes in abundance at family level, (**g**) volcano plots to indicate changes in abundance at species level of selected taxa. Data were obtained from BIOME file of Qime 2.0. L = standard milk; LF = lactose-free milk; BC = blank control; BL = baseline.

**Figure 5 microorganisms-13-02021-f005:**
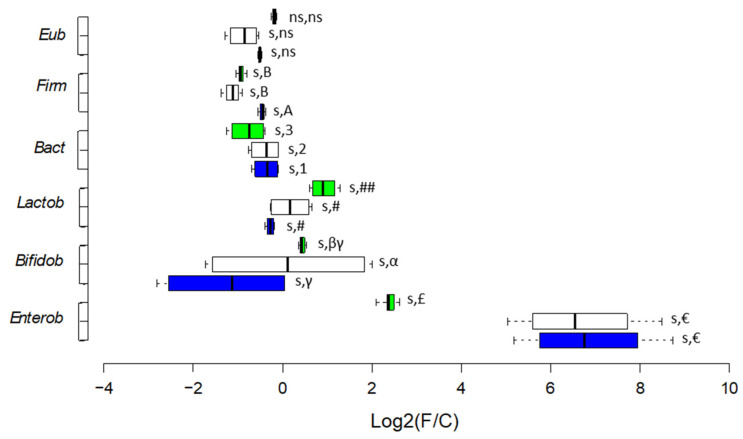
Changes with respect to baseline of fermentation expressed as Log_2_(F/C) of core microbiota taxa measured by qPCR, including data points of 16 h and 24 h during colonic fermentation. Baseline values of absolute abundances, values of shifts, and full statistics are reported in Appendix A. Green = standard milk (L); white = lactose-free milk (LF); blue = blank control (BC). Results are sorted for different taxa and statistical differences are applied for each taxon. s = significant and ns = not significant according to MANOVA and Tukey’s post hoc test for time effect; symbols, letters, and numbers are for MANOVA Tukey’s post hoc test for matrix effect.

**Table 1 microorganisms-13-02021-t001:** Main changes in human colonic microbiota after in vitro fermentation of milk.

Outputs	Lactose Tolerant		Lactose Intolerant [17]	
Metabolomics	UHT Semi-Skimmed Milk (L)	UHT Semi-Skimmed Lactose-Free Milk (LF)	UHT Semi- Skimmed Milk (L)	UHT Semi-Skimmed Lactose-Free Milk (LF)
Organic acids main descriptors	Pentanoic acid and propanoic acid, 2-methyl	Butanoic acid	Pentanoic acid, hexanoic acid, octanoic acid	Butanoic acid
Alcohols main descriptors	2-Octen-1-ol, (E) and 1-Propanol	3-Buten-1-ol, 3-methyl-, benzyl alcohol, phenethyl alcohol, and phenol, 4-methyl	1-Butanol, Phenol	Ethyl alcohol, 1-Octanol, 1-Hexanol, 2-ethyl
Other VOCs main descriptors	Dimethyl trisulfide	Thiazole, 2-methyl	2-Hexanone	2-Acetylthiazole
SCFA production	Increased acetic acid	Unbalanced production	Increased acetic acid	Increased butanoic acid
Detrimental VOCs production	All decreased	Increased Indole	Increased p-cresol	All decreased
**Outputs**	**Lactose tolerant**		**Lactose intolerant** [17]	
**Microbiomics**	**UHT semi-skimmed milk (L)**	**UHT semi-skimmed lactose-free milk (LF)**	**UHT semi-** **skimmed milk (L)**	**UHT semi-skimmed lactose-free milk (LF)**
Metataxonomy (16S-rRNA)	Increased *Bifidobacterium bifidum*	Decreased *Ruminococcaceae;* increased *Veillonellaceae* and *Peptostreptococcaceae*	Increased *Klebsiella* spp.; decreased *Faecalibacterium prausnitsii, Roseburia faecis.*	Unchanged *Verrucomicrobia* phylum; decreased *Peptostreptococcaceae*.
Selected bacterial taxa (qPCR)	Increased *Lactobacillales* and *Bifidobacteriaceae*	Decreased *Lactobacillales* and *Bifidobacteriaceae*	Increased *Lactobacillales* and *Enterobacteriaceae*	Decreased *Bacteroidetes* and *Lactobacillales;* increased *Enterobacteriaceae*.

## Data Availability

Data other than those reported in the MS or in the Appendix A can be requested from the corresponding authors.

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
