# Peer review of "Are “Free From” Foods Risk-Free? Lactose-Free Milk Fermentation Modulates Normal Colon in a Gut Microbiota in Vitro Model"

_microorganisms, 2025, doi:10.3390/microorganisms13092021_

Round 1
Reviewer 1 Report
Comments and Suggestions for Authors
This manuscript investigates the impact of in vitro fermentation of lactose-free milk on normal colon microbiota composition and activity. The study is timely, given the growing popularity of “free from” foods and increasing consumer assumptions regarding their safety and functional equivalence to standard products.
The topic is well aligned with the scope of Microorganisms, and the work is relevant for nutrition science, gut microbiome research, and food product development. The paper is clearly written, methodologically sound, and provides balanced interpretations of the results. Minor clarifications and contextual expansions would improve the manuscript further.
Major comments
- The title is concise, engaging, and accurately reflects the study’s main focus and findings. Consider specifying that the work is an in vitro study to clarify the experimental scope and avoid misinterpretation.
- In the introduction include a short summary of existing studies (if any) on lactose-free dairy fermentation and microbiota modulation to better frame the novelty of this work. Briefly highlight the nutritional and functional differences between lactose-free and conventional dairy products.
- Formulate the hypothesis explicitly to state whether specific microbial or metabolic shifts were expected.
- The in vitro batch fermentation model is appropriate for the research objective, and the choice of endpoints (microbial composition, SCFAs, pH) is suitable. However, some points need clarification:
- Clearly indicate the number of biological replicates (donors) and whether differences between donors were statistically accounted for.
- Provide more details on donor selection criteria (dietary habits, antibiotic use, health status).
- Justify the chosen fermentation time points in relation to physiological relevance.
- Specify whether the lactose-free milk used was commercially available or prepared in the laboratory, and describe its composition.
- Expand the discussion on the functional and potential health implications of the observed microbial shifts.
Minor comments
- The language is clear and professional, but a final editorial check for minor grammatical refinements is recommended.
- Ensure that all abbreviations are spelled out at first mention in both text and figures.
Recommendation:
Minor revision
This is a relevant, clearly presented, and methodologically sound study. The suggested revisions mainly involve clarifying methodological details, enriching contextual discussion, and explicitly noting limitations. Addressing these points will strengthen the manuscript’s clarity, transparency, and scientific value.
Author Response
This manuscript investigates the impact of in vitro fermentation of lactose-free milk on normal colon microbiota composition and activity. The study is timely, given the growing popularity of “free from” foods and increasing consumer assumptions regarding their safety and functional equivalence to standard products.
The topic is well aligned with the scope of Microorganisms, and the work is relevant for nutrition science, gut microbiome research, and food product development. The paper is clearly written, methodologically sound, and provides balanced interpretations of the results. Minor clarifications and contextual expansions would improve the manuscript further.
Major comments
- The title is concise, engaging, and accurately reflects the study’s main focus and findings. Consider specifying that the work is an in vitro study to clarify the experimental scope and avoid misinterpretation.
We thank the reviewer for such comment; the title has been improved so far: “Are “free from” foods risk-free? Lactose-free milk fermentation modulates normal colon on a gut microbiota in vitro model.
- In the introduction include a short summary of existing studies (if any) on lactose-free dairy fermentation and microbiota modulation to better frame the novelty of this work. Briefly highlight the nutritional and functional differences between lactose-free and conventional dairy products.
We are sorry, but we cannot address these two comments. Because, at today there are no new works in literature of lactose free dairy fermentation and microbiota modulation other than that already cited. Additionally, nutritional properties are the same for lactose-free and normal dairy products and differences are just relative to lactose presence (already mentioned), and there is no report on functional differences.
- Formulate the hypothesis explicitly to state whether specific microbial or metabolic shifts were expected.
We thank the reviewer and have added this section at lines 64-68; “We hypothesize that the removal of lactose from dairy products may alter gut microbiota composition in healthy individuals by reducing the availability of lactose-utilizing beneficial bacteria (e.g., Bifidobacterium, Lactobacillus), while potentially favoring the expansion of opportunistic taxa such as Proteobacteria, which can exploit alternative nutrient niches.”
- The in vitro batch fermentation model is appropriate for the research objective, and the choice of endpoints (microbial composition, SCFAs, pH) is suitable. However, some points need clarification:
- Clearly indicate the number of biological replicates (donors) and whether differences between donors were statistically accounted for.
Revised
- Provide more details on donor selection criteria (dietary habits, antibiotic use, health status).
We have added more details
- Justify the chosen fermentation time points in relation to physiological relevance.
We have added this sentence: “Time points were defined as previously described [17-21,23], for example, 24 hours time point is taken to anticipate the beginning of the stationary phase of growth and avoid microbial growth inhibition by toxic microbial metabolites in a batch controlled system”.
- Specify whether the lactose-free milk used was commercially available or prepared in the laboratory, and describe its composition.
This sentence was reported at line 80: “UHT (Ultra High Temperature) semi-skimmed milk (L) and UHT semi-skimmed lactose-free milk (LF) (Granarolo S.p.A., Bologna, Italy”
- Expand the discussion on the functional and potential health implications of the observed microbial shifts.
We thank the reviewer and we added some text in the R1 form 363 to 367.
Minor comments
- The language is clear and professional, but a final editorial check for minor grammatical refinements is recommended.
Revised
- Ensure that all abbreviations are spelled out at first mention in both text and figures.
Reviewer 2 Report
Comments and Suggestions for Authors
In this study, a negative modulation of lactose-tolerant microbiota by fermentation of LF milk was reported, suggesting the functional role of the disaccharide in healthy individuals and possible concerns related to its exclusion. The research falls within the scope of the Microorganisms journal. The topic has certain application value. The research plan and results are relatively reasonable. However, there are still some problems in the writing of the paper that need to be improved.
Specific revision suggestions
1. In the Abstract section, it is recommended to enhance the description of research results and supplement some important research data in the abstract section.
2. In the Keywords section, there is a problem with the writing format of "bifidobacteria". It should be in italics and the first letter should be capitalized.
3. Line34 suggests changing "hydrolysed" to "lactase".
4.Line37-38 "Lactose that is not digested arrives in the colon to be broken down into monosaccharides by gut microbiota", this sentence is not accurate enough. According to existing research, undigested lactose in the colon is fermented by intestinal bacteria into gases (hydrogen, methane and carbon dioxide) and short-chain fatty acids (acetic acid, butyric acid, propionic acid, etc.). Please modify the expression of this part reasonably.
5.In the Introduction section, it is recommended to supplement the research progress related to this study, for example, the current status of in vitro fermentation research on lactose-free milk.
6. According to "2.1.Milk Samples", the sample used in this study is "semi-skimmed milk". It is recommended to explain in an appropriate position in the manuscript the reasons for choosing this sample and why full-fat milk or completely skimmed milk was not used.
7. It is suggested to draw a research flowchart in "2.2. Experimental Workflow".
8. "2.4.In vitro Intestinal Model", this part cites too many literatures. Please consider the rationality of the cited literatures and it is suggested to delete unnecessary literature citations.
9.Figure 2A: It is suggested that this Figure be changed to color, with the same style as Figure 2B.
10.In the "Results" section, there is an issue of unreasonable citation of references. For instance, Line206 has too many references, while Lines 235-307 have no references at all.
11.Conclusions section: There is too much content. The language needs to be concise. Just summarize the research results. There is no need to explain the results. Generally, no references are required for the conclusions.
Author Response
In this study, a negative modulation of lactose-tolerant microbiota by fermentation of LF milk was reported, suggesting the functional role of the disaccharide in healthy individuals and possible concerns related to its exclusion. The research falls within the scope of the Microorganisms journal. The topic has certain application value. The research plan and results are relatively reasonable. However, there are still some problems in the writing of the paper that need to be improved.
Specific revision suggestions
1. In the Abstract section, it is recommended to enhance the description of research results and supplement some important research data in the abstract section.
We thank the reviewer and we have improved the abstract in R1
In the Keywords section, there is a problem with the writing format of "bifidobacteria". It should be in italics and the first letter should be capitalized.
We have used the term bifidobacteria, as a vernacular one for the genus bifidobacterium spp. In this sense it does not go in italics. Altough we have changed the name in the taxonomically proper Bifidobacterium spp.
Line34 suggests changing "hydrolysed" to "lactase".
Sorry we were not able to understand this comment.
4.Line37-38 "Lactose that is not digested arrives in the colon to be broken down into monosaccharides by gut microbiota", this sentence is not accurate enough. According to existing research, undigested lactose in the colon is fermented by intestinal bacteria into gases (hydrogen, methane and carbon dioxide) and short-chain fatty acids (acetic acid, butyric acid, propionic acid, etc.). Please modify the expression of this part reasonably.
We thank the reviewer and we have added the comment in R1
5.In the Introduction section, it is recommended to supplement the research progress related to this study, for example, the current status of in vitro fermentation research on lactose-free milk.
Currently, there are no update in this topic conducted with in vitro models
According to "2.1.Milk Samples", the sample used in this study is "semi-skimmed milk". It is recommended to explain in an appropriate position in the manuscript the reasons for choosing this sample and why full-fat milk or completely skimmed milk was not used.
We chose semi skimmed milk as the most common in Italy and also to have a compromise between the other two products.
It is suggested to draw a research flowchart in "2.2. Experimental Workflow". 8. "2.4.In vitro Intestinal Model", this part cites too many literatures. Please consider the rationality of the cited literatures and it is suggested to delete unnecessary literature citations.
We repute important to cite those reports
9.Figure 2A: It is suggested that this Figure be changed to color, with the same style as Figure 2B.
10.
Figure 2A is coloured in with, black and grey.
In the "Results" section, there is an issue of unreasonable citation of references. For instance, Line206 has too many references, while Lines 235-307 have no references at all.
We apologize and have included reference form 235-307
11.Conclusions section: There is too much content. The language needs to be concise. Just summarize the research results. There is no need to explain the results. Generally, no references are required for the conclusions.
Conclusion section has been improved